# Kidney Cancer and Chronic Kidney Disease: Too Close for Comfort

**DOI:** 10.3390/biomedicines9121761

**Published:** 2021-11-24

**Authors:** Pedro Caetano Pinto, Cindy Rönnau, Martin Burchardt, Ingmar Wolff

**Affiliations:** Department of Urology, University Medical Center Greifswald, 17475 Greifswald, Germany; pedro.pinto@uni-greifswald.de (P.C.P.); cindy.roennau@uni-greifswald.de (C.R.); martin.burchardt@med.uni-greifswald.de (M.B.)

**Keywords:** chronic kidney disease, kidney cancer, hypoxia, new drug modalities, biomarkers

## Abstract

Kidney cancer and chronic kidney disease are two renal pathologies with very different clinical management strategies and therapeutical options. Nonetheless, the cellular and molecular mechanisms underlying both conditions are closely related. Renal physiology is adapted to operate with a limited oxygen supply, making the kidney remarkably equipped to respond to hypoxia. This tightly regulated response mechanism is at the heart of kidney cancer, leading to the onset of malignant cellular phenotypes. Although elusive, the role of hypoxia in chronic kidney diseases is emerging as related to fibrosis, a pivotal factor in decaying renal function. The present review offers a perspective on the common biological traits shared between kidney cancer and chronic kidney disease and the available and prospective therapies for both conditions.

## 1. Introduction

In healthy adults, both kidneys receive about 20–25% of the cardiac output. When considering their relatively small volume and weight, in comparison to other systems, the kidneys are the most perfused organs, receiving a substantially high flow of oxygenated blood [1]. Despite this fact, there is a paradoxical discrepancy between the oxygen levels perfused through the kidney and its actual tissue disposition and consumption. It is estimated that roughly only 10% of the oxygen reaching the kidney is consumed in cellular processes [2]. Arterial oxygen tension (pO_2_) is approximately 100 mmHg (including in the renal artery, leading blood into the kidney) and systemic venous pO_2_ is approximately 30 mmHg. In blood exiting the kidney, via the renal vein, pO_2_ is approximately 70 mmHg. The reason behind this physiological attribute is the peculiar architecture of the renal vasculature [2]. The renal artery and vein branch-out in a parallel pattern, where arterioles and veins are arranged side-by-side, in close proximity [3]. This system allows oxygen to diffuse from the arterioles into the veins with little transit through the capillary network, therefore limiting the concentration of oxygen in the surrounding tissue. The nephrons—the functional units responsible for renal secretion—are exposed to variable levels of pO_2._ In the renal cortex (outer-region) the glomerulus and convoluted tubules experience a higher pO_2_ than the renal medulla (inner-region), approximately 30 and 10 mmHg, respectively. Arguably, the evolution of renal vasculature benefited the secretory functions of the kidneys at the expense of oxygen distribution to the tissue [4].

The kidneys are metabolically demanding organs that require a significant energy output to fulfil their blood filtering and reabsorption roles. Post-glomerular filtration processes depend on an array of membrane-bound transporter proteins, expressed in both the proximal and distal convoluted tubules, responsible for the removal of metabolic bi-products and xenobiotics as well as the reabsorption of solutes, water, glucose, amino-acids, and micro-nutrients [5]. In particular, the renal proximal tubules can concentrate a variety of compounds against steep concentration gradients [6]. Highly specialized renal proximal tubule epithelial cells (RPTEC) remove drugs and toxins from the blood while recovering large concentrations of glucose and sodium from the filtrate back into systemic circulation. RPTEC are rich in mitochondria, responsible for generating the adenosine triphosphate (ATP) necessary to power their active-transport machinery. With a high rate of aerobic respiration and a limited oxygen supply, renal cells constantly operate under potentially precarious pO_2_ conditions [7]. Accordingly, the kidneys are often referred to as hypoxic organs due to their low pO_2_. Hypoxia ensues when O_2_ consumption exceeds supply, and renal cells have developed remarkable adaptations to function with borderline oxygen levels. An intricate regulatory mechanism maintains a fine balance between energy consumption and oxygen supply, preventing the kidneys from falling into an actual hypoxic state, where normal physiological processes can no longer be assured. 

The aim of the present review is to offer a perspective about how the pathophysiological aspects shared by kidney cancer and chronic kidney disease can impact their diagnostics, the development of prospective therapies, and the discovery of novel biomarkers. 

## 2. Regulating the Oxygen Supply to the Kidneys

The central mechanism in the cellular response to fluctuating O_2_ levels is the activity of the Prolyl hydroxylases—Hypoxia-Inducible Factors (PHD-HIF) axis. This interaction functions as a cellular O_2_ sensor, as PHD consume intracellular O_2_ to catalyze the hydroxylation of HIFs, the nuclear transcription factors that regulate gene expression. HIFs activity is kept in check by continuous PHD-mediated hydroxylation [8]. When physiological O_2_ levels are maintained (normoxia), hydroxylated HIFs are targeted for proteolytic degradation, which limits their expression. When O_2_ levels drop, PHD activity is inhibited and HIFs expression is upregulated, prompting a response to counter the effects of reduced O_2_. Several HIFs and PHD isoforms are differentially expressed across renal cells, promoting differential hypoxic responses [9]. A hallmark in renal hypoxic response is the increase of systemic erythropoietin (EPO). This hormone is produced in the fibroblasts of the peritubular interstitium that express PHD2 and stimulates the production of red blood cells, with the objective of increasing the concentration of O_2_ delivered to the kidneys. Glomerular cells respond to hypoxia by releasing vascular endothelial growth factor (VEGF). This growth factor mediates microvasculature growth and repair by stimulating the proliferation of endothelial cells, leading to facilitated blood flow. While these responses are seemingly aimed at restoring renal pO_2_ levels by augmenting supply [8], cells also manage hypoxic events by limiting O_2_ consumption. The activity of adenosine triphosphate (ATP) dependent membrane carriers is reduced and the expression of glycolytic enzymes is enhanced in a push to preclude oxidative phosphorylation in the mitochondria in favor of non-O_2_ mediated anaerobic metabolism ensuring ATP production. These mechanisms, among others, and the fact that they can be readily reversed when physiological O_2_ levels are restored, illustrates the plasticity of renal cells in their hypoxic responses. Beyond the direct role of the PHD-HIF axis in sensing O_2_ levels, this mechanism has a far-reaching impact in cellular regulation. PHD1 and PHD2 suppresses the activity of the nuclear factor-kappa B (NF-kB) pathway, which is involved in cell proliferation and inflammatory responses. PHD3 directly interacts with pyruvate kinase to inhibit glycolytic activity, in a way, by-passing HIF activity. Conversely, HIF activity is also induced independently of O_2_ levels. Post-transcriptional regulation (e.g., phosphorylation) also plays an important role in recruiting the activity of these factors to meet different physiological needs under normoxic conditions. HIF is reported to control the expression of well over 500 genes involved in cell growth, energy production, mobility, angiogenesis, cell cycle, and even gene expression itself (chromatin remodeling) [10]. These pathways are critical to maintain cellular and tissue homeostasis, hence their importance in the hypoxic response.

## 3. Hypoxia Response and Renal Pathophysiology 

The PHD-HIF axis plays a prominent role in renal pathophysiology. Kidney cancer and chronic kidney disease (CKD) are two diametrically opposed pathologies; however, at their core is the regulation of PHD-HIF and associated pathways. Both pathologies share the main, non-hereditary risk factors, including age, high-blood pressure, obesity, diabetes and smoking. Kidney cancer is also a risk factor for renal insufficiency and vice-versa [11]. Renal physiology with its low pO_2_ is susceptible to hypoxic damage, in particular from conditions that compromise blood supply to the kidneys such as vasoconstriction and vascular damage. The resilient nature of kidney cells enables them to adapt and extensively recover from injury events and resume their physiological functions. In particular, the RPTEC, considering their high energy demand, are able to steer their physiology through hypoxic conditions. Mounting evidence underlines the fact that faults in their stress-response machinery are a major contribution to the onset of these renal pathologies (Figure 1).

The most common type of kidney cancer is Renal Cell Carcinoma (RCC). RCC originate from RPTEC that differentiate and acquire a malignant phenotype. Depending on their aggressiveness, RCC can be fast-growing and invasive tumors. This carcinomas are characterized by the loss of function of the von Hippel-Lindau (VHL) protein [12]. This inactivation can result from hereditary factors, sporadic mutations or epigenetic modifications (e.g., DNA methylation), and its end-result is the constitutive activation of HIFs. VHL recognizes hydroxylated-HIF and facilities the activity of the E3 ubiquitin ligase complex, which then mediates HIF proteolytic degradation via ubiquitination. By removing VHL from this mechanism, HIF expression is stable and no longer kept in check leading to the deregulation of its target genes [13]. The balancing act between VHL suppressing and HIF activity is a major determinant in RCC onset, progression and outcome. Two HIF variants, -1α and -2α, are recognized by VHL and are of particular relevance in clear cell RCC (ccRCC), the most common RCC subtype. In the kidneys, HIF-1α is predominantly expressed in tubular cells, while HIF-2α is present in glomerular cells, fibroblasts and endothelial cells. In RPTEC, HIF-1α is key to the regulation of baseline glycolysis and the cell cycle, acting in this way as a tumor suppressing gene. With the onset of the malignant phenotype, cells acquire HIF-2α expression, a factor otherwise absent. HIF-2α is crucial for tumorigenic activity, and it upregulates proliferation, angiogenesis, and mediates inflammatory responses [14]. As cancer cells differentiate further, HIF-1α expression can be completely lost, with HIF-2α assuming its regulatory activity. RCC are highly glycolytic and angiogenic tumors that overexpress glucose membrane transporters and VEGF to meet the needs of their anaerobic metabolism [15]. 

Epidemiological studies show an interdependent link between CKD and the development of urogenital cancers, both direct and indirect. Early stages of renal insufficiency are not correlated with cancer onset, however late stage CKD patients have a 10–20-fold higher cancer incidence. This is understood to be derived from the systemic accumulation of toxic metabolites and, eventually, pharmaceuticals after diminished renal function, a factor that leads to both cellular toxicity and an impaired immune response [16]. On the other hand, several chemotherapeutical agents can induce kidney damage, given their cytotoxic nature and renal excretion. The latest generation of anti-cancer drugs (e.g., tyrosine kinase inhibitors, biopharmaceuticals) dramatically reduced nephrotoxic effects, overcoming chemical-induced renal damage associated with drugs of preceding generations, cisplatin being the classical example. Partial nephrectomy, the surgical procedure to remove an RCC tumor, frequently requires clamping of the renal vessels, effectively interrupting blood flow. This iatrogenic ischemia can also result in hypoxic damage to the kidney. There is a positive association between RCC diagnosis and CKD, however any underlying mechanisms connecting both pathologies are still largely unknown [11].

While the role of the PDH-VHL-HIF axis is well-characterized in RCC, its impact and the role played by hypoxia in CKD is far more elusive. CKD is characterized by the loss of overall kidney function and is a complex, multifactorial and insidious disease. It often leads to renal failure over time without appropriate clinical management [17]. Contrary to RCC, this pathology virtually affects all cell types in the nephron and not exclusively RPTEC. The effects of hypoxia in CKD are traditionally assumed to follow detrimental damage to the renal capillary network [18]. A restricted blood flow leads to a chain reaction where O_2_ deprived renal cells promote scaring of the peritubular space and irreparable damage to nephrons and the vasculature. This damage is derived from renal fibrosis, one of the most prominent factors contributing to CKD pathophysiology, characterized by the deposition of extracellular matrix (ECM) proteins in the peritubular space. In an O_2_ deficient environment, the turn-over of RPTEC is compromised, and aging epithelial cells result in the loss of tubules and degrading renal function [19]. The understanding of the impact of hypoxia in CDK is challenged by the contradictory activities of HIF-1α, described across a multitude of comprehensive studies. On one hand, HIF-1α activity facilitates the recovery of damaged tubular cells by controlling their de-differentiation and growth, suppresses inflammation, fibrosis and improves renal function. On the other, HIF-1α seemingly promotes the opposite, increasing fibrosis and accelerating tubular and glomerular damage. Interestingly, most detrimental effects were observed in studies involving the overexpression or knock-out of HIF-1α, while most positive effects were observed in studies using pharmacological interventions. This evidence is detailed by Faivre et al. [20], and arguably the experimental models used may be an important factor in the HIF-1α activity reported. Nonetheless, these dual effects suggest that HIF-1α post-transcriptional regulation, rather than its expression, dictate its physiological impact. Dedifferentiating renal epithelial cells can acquire a proto-fibrotic phenotype and drive tubulointerstitial inflammation [8]. HIF-1α appears to drive both regeneration and epithelial to mesenchymal transition (EMT) prompting the proliferation of differentiating RPTEC that did not fully recover their epithelial phenotype. These proto-fibrotic cells deposit extracellular matrix and can acerbate functional losses by promoting tissue fibrosis [21]. In a similar fashion, interstitial fibroblasts, in the peritubular space, can proliferate and deposit ECM when pushed towards a predominantly anaerobic metabolism. This process is mediated hypoxia-independently, with the transforming growth factor beta (TGF-β), a cytokine that controls cellular proliferation, inhibiting the activity of PHD2 resulting in unbalanced HIF expression [22]. The role of inflammation as another hypoxia-independent activator of HIF in pathophysiological conditions is now emerging, with different regulatory pathways interacting with HIF. The tumor necrosis factor alpha (TNF-α) is a cytokine released by macrophages in response to cellular stress, and can indirectly stabilize the cellular levels of HIF-1α via the transcriptional activity of the NF-kB. 

## 4. Common Traits in RCC and CKD

Contrary to RCC, no evidence suggests that VHL expression and activity is compromised throughout the onset and progression of CKD. This supports the fact that a key difference between both pathologies is that HIF activity does not go completely unchecked during CKD. RCC, particularly in advanced stages, are highly inflammatory tumors. RCC release an array of cytokines believed to contribute to the maintenance of its microenvironment and the self-regulation of its cancer phenotype. Interleukin-6 (IL-6) and TNF-α play major roles in cancer proliferation by regulating cell growth and metabolism. Recently, systemic inflammation has been proposed as a marker for RCC progression [23]. The secretion of cancer-specific cytokines and chemokines enables advanced RCC to sequester the activity of immune cells to facilitate a metastatic cascade, leading to the spread and engraftment of cancer cells outside of the primary tumor [24]. The inflammatory nature of RCC can be considered as a mechanism responsible for cancer survival and proliferation, with tangible effects in systemic inflammation. Intra-tumor fibrosis (ITF), is the result of a complex interaction between cancer and infiltrating cells that results in dense ECM deposits populating primary tumors and plays an important role in maintaining the cancer microenvironment and as a repository of immune cells [25]. Circumstantial evidence shows an association between ITF and the progression of RCC into invasive tumors with poor clinical prognosis; however, to date little is known about the role that ITF plays in RCC pathophysiology [26]. The same mechanisms involved in renal fibrosis (e.g., TGF-β, EMT) are present in ITF tissue, and a better understanding of how prominent fibrosis is on RCC onset could shed light on the common causes of both diseases before they evolve and develop their intrinsic phenotypes. Moreover, the impact that both inflammation and ITF have on normal renal physiology is uncertain.

## 5. Available Therapies for RCC and CKD

Clinically, both pathologies present diverse outcomes. When detected at an early stage, RCC treatments offer a good prognosis. Tumor resection is the front-line treatment for localized RCC, enabling the removal of the tumor with limited impact on kidney function. Advanced RCC are associated with a poorer prognosis and about a third of RCC patients are estimated to be diagnosed with metastatic tumors. CKD is a progressive disease with very limited treatment options. About one in ten adults globally is estimated to experience some form of CKD. Managing lifestyle by controlling diet and blood-pressure -as well as maintaining physically active remain the best options to minimize the effects and slow the decay of renal function. Nonetheless, CKD often leads to renal failure, requiring renal replacement therapy, begins with dialysis and eventually leads to kidney transplant. RCC is stealthy and challenging to diagnose given its lack of symptoms during initial stages. This pathology does not seem to interfere with normal kidney function, even in a mid to advanced stage, rendering common markers to evaluate renal function, such as glomerular filtration rate (GFR), anemia and decreased systemic sodium, ineffective for its detection. Most RCC cases are detected by chance, when patients undergo diagnostic imaging (e.g., ultrasound, tomography scan) for unrelated reasons. Frequently, tumors are detected in patients in a risk group for renal insufficiency (e.g., high blood pressure, diabetes) that are tested for renal damage. Over the past two decades the treatment of RCC, namely in its advanced stages, saw a dramatic improvement. Patients with previously poor clinical prognosis have benefited from the introduction of, mainly, two novel drug classes, immune check-point inhibitors (ICI) and multi tyrosine kinase inhibitors (TKI). ICI are large biological immune-therapy molecules, either whole antibodies or antigen-binding fragments (Fab), that block the binding of check-point receptors in T-cells to their respective membrane ligands. Check-point receptors regulate immune responses and, under normal physiological conditions, help T-cells to discriminate between autologous, healthy and foreign cells, preventing disproportionate immune cascades. RCC cells shield themselves from the immune system by presenting specific check-point ligands in their membrane. ICI facilitate the activation of T-cells by exposing RCC as disease tissue, inducing cellular death pathways in tumor cells. TKI are small-molecule drugs that target the activity of specific tyrosine kinase proteins that regulate key cellular processes. TKIs effective in the treatment of RCC mainly target different subtypes of VEGF receptors and block their angiogenic activity. These drugs prevent endothelial cells in the vasculature from responding to VEGF secreted by RCC, compromising the tumor’s angiogenic activity and blood supply and therefore hindering cancer proliferation. Additionally, inhibitors of the mammalian target of rapamycin (mTOR) are also used to treat RCC [27]. The mTOR pathway is an upstream regulator of VEGF synthesis and plays a central role in cell proliferation and differentiation [28]. Its inhibition blocks VEGF release and hampers the proliferation of cancer cells, hence the benefits of mTOR inhibitors in RCC. Current front-line pharmacological interventions to treat RCC, depending on disease severity and risk factors at the time of diagnosis, consist of therapies combining ICI and TKI or different ICI molecules (Nivolumab and Ipilimumab) [29]. As of 2021, there are about 15 molecules approved for the treatment of RCC as single agents or within combinations by the European Medicines Agency (EMA) and the Federal Drug Administration (FDA) [30,31].

Contrasting with this scenario, the first molecule to treat the progression of CKD was FDA, which was approved in 2021. Dapagliflozin is a sodium glucose transport protein 2 (SLGT2) inhibitor, developed (and approved) to treat type 2 diabetes (T2D). It lowers blood sugar levels by preventing the RPTEC in the kidney from reabsorbing filtered glucose. This drug was repurposed to treat CKD after it substantially reduced the risk of renal failure and the onset of end-stage renal disease in patients with or without T2D [32]. In addition to blocking glucose uptake, SLGT2 inhibition in RPTEC reduces sodium uptake, which in turn reduces the workload of the Sodium-Potassium-ATPase (Na/K-ATPase) efflux pump. Na/K-ATPase are highly expressed in RPTEC and central to their physiology (e.g., maintaining electrochemical gradients, concerted activity with SLGT2, osmotic balance) and it is estimated that these pumps consume over a third of the cellular ATP production [33]. The downregulation of Na/K-ATPase activity is seemingly beneficial to renal physiology since it minimizes energy demand and incidentally reduces O_2_ cellular consumption. With diminished pressure on their O_2_ supply, HIF physiological regulation is restored and cells become more resilient to hypoxic events. Additionally, lower secretion of sodium into the renal medulla alleviates vasoconstriction. This reduces the stress in endothelial cells, improves their function and minimizes vascular damage, while restoring renal O_2_ supply [34]. Dapagliflozin represents a direct pharmacological intervention to counter the progression of CKD; nonetheless, other therapies destined to manage other conditions can additionally help to prevent renal damage and the onset of CKD. Anti-hypertension drugs may exert a long-term protective effect in the kidneys by minimizing the effects of high blood pressure and facilitating vasodilation [35]. Moreover, evidence also shows that SLGT2 inhibitors have a cardioprotective role, given their anti-inflammatory and anti-fibrotic effects in cardiomyocytes derived from lower intracellular sodium levels [36]. Allegedly, one of the reasons behind the very limited therapeutic options for CKD is our limited understanding of its pathophysiology. This is compounded by the fact that kidney function decays with age, affecting the pharmacodynamics of several therapeutical agents and therefore precluding their potential use in the treatment of CKD [37]. 

## 6. Prospective New Therapies for RCC

RCC, despite its complexity, has at its heart a phenotypical de-differentiation driven by the PDH-VHL-HIF axis that leads to high angiogenic activity. This fact, together with evidence from extended cancer research, enabled the development of targeted therapies, not only for RCC but for cancers that share similar regulatory pathways [38,39]. Moreover, the vast majority of drugs and treatments currently under development for RCC are focused on novel ICI and TKI molecules and the combination of approved drugs, respectively [34,35]. This illustrates the efficacy of the current therapies available. Noteworthy are ongoing trials to investigate the effects of combining ICI with experimental cancer vaccines (NCT02950766). The objective of this new approach is to improve the efficacy of immunotherapy including in patients with a limited response to treatments [40]. Mutations in developing tumors create neoantigens, a type of proteins that are unique to individual cancers. Contrary to conserved immune-histocompatibility antigens that are shared amongst human populations, neoantigens are recognized by T-cells as foreign entities and can trigger an immune response [41]. However, factors such as limited immune cell infiltration, tumor-derived T-cell inhibition and a high neoantigen turn-over restrict the response under normal circumstances. Evidence from pre-clinical studies shows that mRNA-based vaccines encoding neoantigens substantially amplify T-cell mediated tumor cell death and maintain an adaptive immune response [42]. This effect is achieved thanks to the priming of a substantial pool of T-cells to recognize neoantigen expressing tumor cells and memory B-cells. Given the unique nature of neoantigens the clinical trial for this novel therapy requires a personalized medicine strategy where tumor samples are analyzed and target neoantigens sequenced to produce a patient-specific mRNA vaccine [43]. RCC is a promising cancer type for ICI—mRNA vaccine combination, considering their relatively stable mutation rate and a high proportion of neoantigens [44]. The rationale behind this strategy is to amplify and sustain the ICI immune response by activating a substantially higher numbers of T-cells to target tumors. This can potentially overcome poor ICI response in certain patients (e.g., with depressed immune systems) by zooming T-cells directly to the tumors, bypassing the tumors immune-suppressive microenvironment. One of the challenges facing this new approach is tumor heterogenicity and the chance of neoantigen depletion, where only tumor populations cells with a specific set of neoantigens are targeted leaving others, that have mutated, unscathed to proliferate [45]. Tools that can predict neoantigen sequences and vaccines incorporating multiple targets can overcome these issues, as well as early detection of RCC, minimizing the consequences of an aggressive cancer phenotype [46]. As of 2021, current phase I clinical trials are in the recruiting stage and expected to yield results in the coming years. With no RNA-based vaccines yet approved for cancer treatments as well as the challenges of developing and producing patient-specific molecules in the timeframe required for an effective therapy, real-world applications of ICI-mRNA vaccines are, arguably, still several years away. Nonetheless, this innovative therapy is set to represent another substantial improvement in the treatment of RCC patients. 

## 7. Prospective New Therapies for CKD

Extensive research into CKD pathophysiology has revealed promising therapeutic targets for CKD that could effectively treat the disease, beyond mitigating its onset and progression. The dissemination of *omics* techniques and the advent of novel bioinformatic analytical tools that can scout over large metadata sets are also contributing to the identification of candidate CKD druggable targets [47,48]. Renal inflammation and fibrosis mechanisms are the focus of new therapeutic approaches, unsurprisingly, given their recognized role in the progression of CKD [49]. Bardoxolone is an inducer of the Nuclear factor erythroid 2-related factor 2 (NRF2) and was a promising therapeutical agent [50]. Different clinical trials reported that patients diagnosed with diabetic nephropathy, late stage CKD [51], glomerulosclerosis and polycystic kidney disease had a significant recovery in GFR when treated with bardoxolone [52]. NRF2 is a transcription factor central to several regulatory mechanisms in mice models in which this factor is dysregulated or knocked-out; the animals developed glomerular damage and fibrosis leading to the loss of renal function [53]. NRF2 keeps inflammatory responses in check by mediating the expression of anti-oxidant proteins that preserve cellular redox balance and maintain cellular homeostasis [54]. Despite its benefits, cardiovascular safety issues hampered the clinical implementation of bardoxolone [49]. Nonetheless, this drug showed that targeting inflammatory cytokine release is a viable clinical avenue to treat CKD. Strategies to treat renal fibrosis have targeted TGF-β inhibition [55]. These efforts aim to prevent phenotypic changes in renal cells by blocking proliferation and differentiation pathways, namely preventing RPTEC from becoming proto-fibrotic. Pirfenidone is a growth factor inhibitor used to treat pulmonary fibrosis, and it has been reported to also block TGF-β activity [56,57]. Previous trials did not determine a positive effect of pirfenidone in GFR loss. Nonetheless, an ongoing clinical trial (NCT04258397) is investigating the effects of this drug in preventing the development of renal fibrosis using comprehensive methodology (e.g., Imaging, renal function biomarkers). Neutralizing monoclonal antibodies are highly specific drugs that minimize off-effects and can efficiently block TGF-β activity [58]. Human trials of these molecules did not meet their efficacy endpoints, with no significant improvement in GFR after treatment, despite promising pre-clinical results. This was attributed to the tissue distribution properties of the antibodies and limited delivery to the kidneys and affected organ areas [59]. Direct targeting of TGF-β continues to be a promising therapeutic strategy, providing that the pharmacokinetic issues surrounding renal delivery of biological drugs are addressed. 

Commonly prescribed medications are also being sought as potential CKD treatments. The inflammatory process stimulates Angiotensin II (AII) in a positive feedback loop. Therefore, reducing AII availability negatively impacts the release of inflammatory factors [60]. Angiotensin-converting enzyme inhibitors (ACE) effectively reduced renal inflammation and fibrosis, independently of blood-pressure, in pre-clinical models [61]. Clinically, and across multiple studies, ACE reduced the levels of systemic inflammatory factors and pro-inflammatory monocytes, and have overall renoprotective effects, ameliorating disease progression and reducing the risk of dialysis in CKD patients [62]. Despite their benefits, ACE are, so far, recommended as a preventative medication for CKD in patients with pre-existing conditions and not necessarily as a treatment. 

## 8. Perspective on the Use of New Drugs Modalities in RCC and CKD

Experimental molecules with a biological activity, physical-chemical properties and pharmacokinetics substantially different from conventional molecules—commonly known as small-molecules—and are often referred to, unofficially, as new drug modalities (NDM) [63]. NDM include a very diverse collection of molecules and have drawn much attention from drug makers given their potentially very high efficacy and marginal toxicity [64]. 

A prominent class of NDM are RNA based drugs; which can be designed to upregulate or block the expression of target proteins [65]. The aforementioned mRNA vaccines targeting RCC can be considered a type of RNA drug. Anti-sense oligonucleotides (ASO) are highly-stable, short, single-stranded oligonucleotide sequences that bind mRNA and prevent protein translation. ASO can be tailored to virtually block the expression of a protein of interest [66]. These molecules are hampered by limited distribution and tend to accumulate in tissues while having very slow elimination rates. Interestingly, ASO accumulate in the kidneys in large amounts, mainly in the proximal tubules, via an endocytic mechanism not yet fully understood [67]. Therefore, ASO have been considered as a vehicle to easily reach therapeutic targets in RPTEC, and the latest generation of ASO has shown remarkable renal safety [68]. Autosomal polycystic Kidney Disease (APKD), is a hereditary condition characterized by the formation of large fluid-filled renal cysts, and leads to renal failure [69]. Its genetic roots and molecular mechanisms of disease are reasonably understood [70]. ASO directed at the mTOR complex normalized the kidney function while reducing weight and cyst size in an orthologous mice model for APKD [71]. Alport syndrome (AS) represents another hereditary disease leading to renal failure and is characterized by deficient collagen IV, which results in glomerulonephritis [72]. ASO designed to truncate the expression of the COL4A5 gene (collagen IV alpha-5-chain) successfully improved survival in male animal models of X-linked AS [73]. In the kidney cancer field, ASO directed at VEGF successfully promoted the remission of tumors in RCC xenografts [74]. While pre-clinical data is promising, ASO development is complex and requires the identification of highly specific genetic targets. Despite advances in ASO safety, concerns remain about the pathological effects of the long-term accumulation of ASO in RPTEC [75], a main reason behind the slow pace of clinical ASO research. In the search for a CKD target, genetic variants of apolipoprotein L1 (APOL1) have been extensively associated with the development of CKD [76]. APOL1 expression leads to the loss of podocytes and, incidentally, renal function, and a recently initiated phase I trial (NCT04269031) is set to start evaluating the potential of anti-APOL1 ASO in treating CKD [77]. 

Anticalins are genetically modified lipocalins, a family of small human binding proteins, and represent artificial proteins that act as antibody mimetics [78]. An advantage of these synthetic peptides is their small size relative to conventional monoclonal antibodies (about 1/8 of the size). This fact explains anticalins’ improved tissue penetration properties, benefiting drug delivery and better clearance properties and minimizing potential side-effects due to prolonged exposures [79]. In a first-in-human trial involving RCC patients, an anti-VEGF anticalin effectively rendered VEGF undetectable in systemic circulation [80]. Ensuing pre-clinical studies demonstrated that this anticalin inhibits the VEGF-mediated proliferation of endothelial cells, reduces micro vessel density, and improves vascular permeability. Relative to Bevacizumab, an approved anti-VEGF monoclonal antibody, anticalins show improved safety with no platelet aggregation or thrombus formation observed [81]. 

Proteolysis Targeting Chimeras (PROTAC) are bifunctional molecules that mediate selective protein degradation [82]. PROTAC consist of two protein binding domains linked together; one domain interacts with E3 ubiquitin ligases and the other with any given protein [83]. Therefore, PROTAC are designed to bind a protein of interest and target it for degradation by activating the ubiquitin-proteasome system. PROTAC are highly specific and efficient in knocking-out their target proteins and, relative to other new modality drugs, their pharmacokinetic properties are closer to those of small molecules, and do not tend to accumulate in tissues over time [84]. The interest in these molecules is gaining momentum in oncology research, considering their potential ability to remove upregulated or abnormal proteins responsible for malignant phenotypes [85]. PROTAC research is still in its infancy, with the first E3 ligase chimera reported in 2008 [86]. Nonetheless, the opportunity to develop therapies for common diseases lacking effective treatments has driven the interest in PROTACS [87]. This is illustrated by the development of ARV-110, an androgen receptor degrader currently undergoing phase II trials (NCT03888612) for the treatment of castration resistant prostate cancer [88]. There is limited research in the PROTACS applications to tackle renal pathologies. Noteworthy is the development of VHL-recruiting PROTACS [89]. Although VHL-degraders are likely ineffective in RCC with VHL loss of function [90], they could potentially bypass VHL activity and target HIF directly to ubiquitin ligases. On the other hand, small molecule inhibiters of the VHL-HIF interaction, such as VH298, have been developed. These molecules mimic hypoxia, stabilize HIF activity [91], and represent a step forward in our toolbox to better understand and develop pharmacological interventions to treat renal hypoxia, which represent the common root of RCC and CKD. 

## 9. Biomarkers in RCC and CKD Early-Diagnosis

Beyond the development of innovative therapies, the identification of early biomarkers can substantially improve the clinical management of both pathologies. Early diagnosis of RCC will enable the timely implementation of treatment, improving patient prognosis as well as decreasing the risk of cancer associated CKD and vice versa [16]. In recent years, two different classes of biological molecules have emerged as active players in both RCC and CKD pathophysiology and have been proposed as potential biomarkers. MicroRNA (miRNA) are ubiquitous single-stranded non-coding RNA molecules that can assume multiple regulatory functions, namely acting as post-transcriptional regulators. With more than 2500 human miRNA identified [92], several have been characterized as extracellular miRNA, found in circulation or in body fluids (e.g., urine) in both healthy and diseased conditions. It has long been postulated that identifying a unique miRNA signature will help not only to diagnose a disease, but to provide information about its severity. miR-21 expression was found to be induced by TGF-β and correlated with an increase in renal fibrosis in pre-clinical CKD models and patients experiencing renal failure [93]. Concurrently, miR-21 induction, in vitro, resulted in the upregulation of fibrotic and inflammatory mediators including TGF-β and IL-6 [94]. Fibrosis and the deposition of ECM components (e.g., collagen I, fibronectin) is also associated with the activity of miR-433 and miR-484. On the other hand, deteriorating renal function and the onset of fibrosis is also correlated with the downregulation of certain miRNA. Decreased miR-29c expression is associated with interstitial fibrosis and, interestingly, HIF-1α activation restores miR-29c levels, leading to ameliorated fibrotic conditions [95]. Characterization studies have shed some light on the miRNA profiles across different conditions leading to renal failure. miR-1, miR-133 and miR-223/miR-199 were found to be upregulated only in the urine of APKD patients relative to CKD patients, suggesting a specific profile for these conditions [96]. In a CKD patient cohort, miR-21 serum levels were consistently increased relative to healthy controls [97]. The identification of these profiles may prove beneficial in monitoring disease progression and treatment response rather than as early biomarkers, since they reflect conditions where the loss of kidney function is already substantial. 

In a similar fashion, several miRNAs have been associated with RCC. Several studies have identified a substantial number of deregulated miRNA, a common trend in cancer, as miRNA play, arguably, an important role in tumor homeostasis [98]. Deregulated miRNA in RCC are reported to impact genes involved in the regulation of cytokines, placing them as mediators of RCC immunogenicity [99]. miR-21 seems to be a promising candidate for a diagnostic use in RCC and for disease monitoring. The expression of miR-21 is reported as elevated in RCC patient serum and found to be decreased after tumor resection [100]. Most interestingly, miR-21 is equally elevated in CKD patients and associated with fibrosis, while in RCC it has been linked to tumor invasion and angiogenesis [101]. An in depth understanding of miR-21 regulation will help us to understand the role of fibrosis in both CKD and RCC and clarify whether its presence in CKD patients represents a hidden sign of tumorigenesis. Other detectable miRNAs reportedly involved in RCC include miR-200a, which in early stage RCC is elevated in serum and decreased in urine [102], and miR-150, which has serum levels that correlate with patient survival [103]. Overall, miRNA associated with RCC may prove to be a tool to monitor treatment and disease progression. However, their implementation as diagnosis biomarkers still requires a significant understanding of their functional roles and profiles in both heathy conditions and disease. 

The neutrophil gelatinase-associated lipocalin (NGAL) is a recognized biomarker for CKD [104,105]. Levels of this low-weight protein are increased in the urine of patients with renal injury, and are reported to reflect the severity of the disease [106]. This biomarker is used not only for diagnosis but also to evaluate CKD progression. NGAL acts as mediator of the innate immune response to bacterial infections, and its expression is induced by TNFα and IL-6 [107]. It is shed from RPTEC in response to inflammation [108]. On the other hand, little is known about the role of NGAL in RCC. Different studies have determined that NGAL expression is upregulated in kidney cancer tissue at the mRNA and protein levels [109,110]. However, the significance of NGAL levels in RCC patients is currently unresolved [111], and further research in the role of NGAL in RCC pathophysiology may elucidate the future impact of this biomarker in kidney cancer. Interestingly, a study shows that NGAL levels in the serum of RCC patients treated with sunitinib have a strong correlation with progression-free survival [112]. This suggests that the serum levels of NGAL have the potential to monitor RCC treatment outcomes, as opposed to its urine concentration, which is used in the follow-up of patients with CKD. 

Thioredoxins (TRX) are a family of ubiquitous proteins that maintain the cellular redox balance and act as molecular switches in multiple regulatory pathways [113]. The cellular redox balance has long been recognized as an import factor in disease. An increase in the oxidation status of molecules is associated with damage to the genetic material and cell death. O_2_ easily sequesters electrons (e-), acting as a potent oxidizer. To counter this effect, TRX act as e- donors and help to restore the redox balance. Thanks to the oxidative nature of aerobic respiration, the cellular redox state is tightly controlled, and beyond transferring e- to oxidized proteins, TRX regulate cellular energetics by directly controlling glucose uptake and glycolysis. The inhibition of Thioredoxin 1 (Trx1) results in increased glycolytic activity, while suppression of the Glutaredoxin 1 (Grx1) yields the opposite effect [114], and the activity of Thioredoxin-interacting protein (TXNIP) down-regulates the expression of the glucose uptake transporter 1 (GLUT1), impacting the intracellular availability of glucose. Clinical association studies determined that a reduced TXNIP expression in RCC corelates to a poor survival outcome [115]. TRX are believed to play a role in remote-sensing and cell-to-cell communication. In diabetic patients, Trx1 and TXNIP are found in urine, however, TXNIP is only found in cases where renal function is declining [116]. TRX are believed to play a role in remote-sensing and cell-to-cell communication; they can be found in blood and urine, although their secretory mechanism remains unresolved. Extensive research is still required to understand the role of TRX in hypoxia and renal disease, nonetheless the characterization of secreted TRX profiles in CKD and RCC patients, similarly to miRNAs, may prove a valuable tool to discriminate between both pathologies at an early stage. 

## 10. Conclusions

While the introduction of targeted and immune-therapies dramatically improved the treatment of RCC [117], there has yet to be a major development in CKD therapy. Nonetheless, the next generation of pharmaceuticals may deliver such a breakthrough. A better and more comprehensive understanding of renal disease pathophysiology is driving the identification of new therapeutic targets and contributing to the unravelling of the DMPK—disposition, metabolism and pharmacokinetics—, safety and efficacy of novel, less permeable and more chemically stable molecules. The genetic and molecular mechanisms governing pathologies such as APKD, AS or even RCC are currently fairly understood, a factor that dramatically contributes to the design and development of specific therapies. The heterogenicity of CKD, which effectively represents a collection of different renal conditions, has posed a challenge in unravelling a druggable target. While the role of fibrosis in CKD is now evident, its contribution to the onset and progression of RCC remains elusive. On one hand, tackling fibrosis in CKD may improve renal function and prevent its decay by preserving tubular structures. On the other hand, fibrosis is likely involved in the RCC self-preserving inflammatory firewall, and reducing it may expose tumors and facilitate the activity of immunotherapy. Identifying common factors and regulatory pathways in CKD and RCC, such as hypoxic response regulation, is a step towards the discovery of biomarkers that can be used to diagnose and discriminate both pathologies at an early stage. 

## Figures and Tables

**Figure 1 biomedicines-09-01761-f001:**
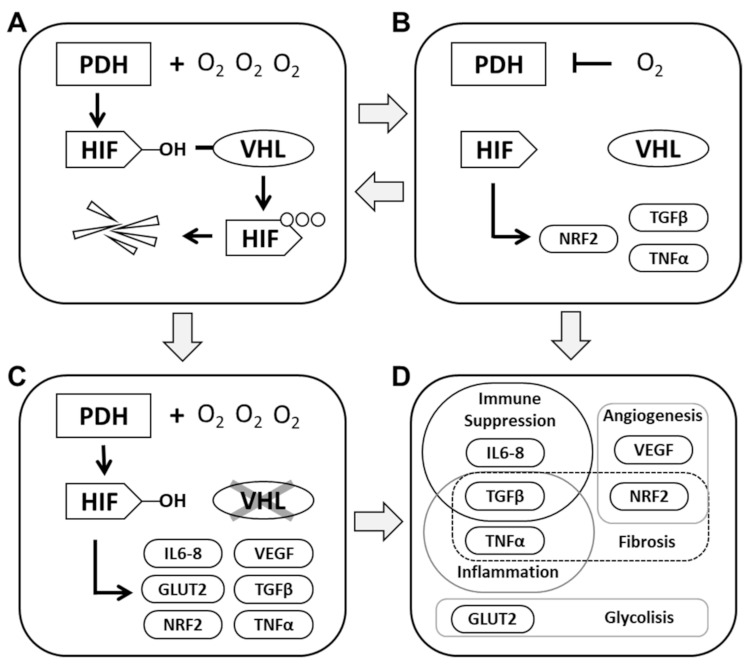
Cellular response to oxygen levels. (**A**): Under normal conditions, PHD have access to sufficient oxygen levels to promote the hydroxylation of HIF and maintain a stable expression of these transcription factors. Excess HIF is a target for proteolytic degradation mediated by VHL. (**B**): When cellular oxygen levels drop below the levels required to ensure PDH activity, HIF expression is destabilized. VHL is precluded from recognizing HIF and a lack of degradation leads to the activation of a myriad of genes with diverse functionalities. HIF activity will ensure cell survival and facilitate the restoration of physiological oxygen levels. Unchecked HIF activity can result in the sustained expression of inflammatory factors. (**C**): In RCC, the loss of VHL activity leads to the constitutive activation of HIF and a predominantly inflammatory and unbalanced cellular activity. (**D**): The differential activity of HIF in low oxygen conditions or in the absence of VHL leads to upregulation of several interconnected cellular pathways.

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
