# Peer review of "Kidney Cancer and Chronic Kidney Disease: Too Close for Comfort"

_biomedicines, 2021, doi:10.3390/biomedicines9121761_

Round 1

Reviewer 1 Report

The paper “Kidney Cancer and Chronic Kidney Disease: Too Close for Comfort" by Pinto et al.is a review on Kidney cancer and chronic kidney disease.

 The article is well written, and only minor spell check is needed. The study has a good design. The article is logically divided into sections and subsections. There is one figure of good quality. The references cited are relevant and adequate. The work has an average degree of novelty and of good interest to the readers.

Comments:

  • Line 253: please use lifestyle instead of live-style
  • SGLT2i beneficial effects on kidney should be improved by adding the beneficial anti-inflammatory and anti-fibrotic effect of this class drug (doi: 10.3390/ijms22115863), as well as their role on endothelial disfunction (doi: 10.3390/biomedicines9101356)
  • Paragraph 5, line 310-311: It is true that one of the reasons behind the very limited therapeutic options for CKD is our limited understanding of its pathophysiology. However, a great impact is also played by the decay in kidney function which makes unavailable most drugs (10.3390/medicina55100617).

Author Response

Reviewer 1 comments and suggestions – rebuttal

Reply: We want to thank the reviewer for the constructive comments to our manuscript entitled: ´´ Kidney Cancer and Chronic Kidney Disease: Too Close for Comfort´´. In particular, the suggestions regarding the impact of SLGT2 inhibitors in inflammation, fibrosis and endothelial function are highly appreciated. We have updated our manuscript to reflect the reviewer´s considerations and we hope our corrections are satisfactory.

The paper “Kidney Cancer and Chronic Kidney Disease: Too Close for Comfort" by Pinto et al.is a review on Kidney cancer and chronic kidney disease.

The article is well written, and only minor spell check is needed. The study has a good design. The article is logically divided into sections and subsections. There is one figure of good quality. The references cited are relevant and adequate. The work has an average degree of novelty and of good interest to the readers.

Comments and suggestions

  • Line 253: please use lifestyle instead of live-style

Reply: live-style replaced by lifestyle

  • SGLT2i beneficial effects on kidney should be improved by adding the beneficial anti-inflammatory and anti-fibrotic effect of this class drug (doi: 10.3390/ijms22115863), as well as their role on endothelial disfunction (doi: 10.3390/biomedicines9101356)

Reply: The benefits of SLGT2 inhibitors to improve endothelial function are now stated in the manuscript. Salvatore et al 2021 was added as a reference. The anti-inflammatory and anti-fibrotic effects of SLGT2 inhibitors are now stated in the manuscript. Palmiero et al 2021 was also added as a reference (lines 305-308 and 312-314).

  • Paragraph 5, line 310-311: It is true that one of the reasons behind the very limited therapeutic options for CKD is our limited understanding of its pathophysiology. However, a great impact is also played by the decay in kidney function which makes unavailable most drugs (10.3390/medicina55100617).

Reply: The impact of decaying renal function in precluding the potential use of therapeutical agents to treat CKD is now stated in our manuscript. Caturano et al 2019 was added as a reference (lines 316-318).

Reviewer 2 Report

Reviewer comments and suggestions

The manuscript “Kidney Cancer and Chronic Kidney Disease: Too Close for 2 Comfort. The present review offers a perspective on the common biological traits shared between kidney cancer and chronic kidney disease and the available and prospective therapies for both conditions. Although I didn’t find any novelty in this review. In my opinion, this manuscript needs more insights information regarding kidney and chronic kidney disease and prospective therapies with molecular mechanism. Please see my comments and suggestions.

Comments and suggestions

  1. Keywords or keywords please check and confirm. And each keywords is small letter not capital.
  2. Post-glomerular filtration processes depend on an array of membrane-bound transporter proteins, expressed in both 41 the proximal and distal convoluted tubules, responsible for the removal of metabolic bi products and xenobiotics as well as the reabsorption of solutes, water, glucose, amino acids, and micro-nutrients. This sentence has written without citation.

  1. At the end of introduction you have to write aim of your current study that is missing.

  1. Section 2

The central mechanism in the cellular response to fluctuating O2 levels is the activity of the PHD-HIF axis. This interaction functions as a cellular O2 sensor, where Prolylhy droxylases PHD consume intracellular O2 to catalyze the hydroxylation of nuclear 61 transcription factors known as Hypoxia-Inducible Factors HIFs. What is the meaning of this sentence? Please check and confirm. PHD-HIF if first time you have to must abbreviate it.

  1. The activity of ATP-dependent membrane carriers is reduced and the expression of glycolytic enzymes is enhanced, in a push to preclude oxidative phosphorylation in the mitochondria in favor of non-O2 mediated an-78 aerobic metabolism ensuring ATP production. If you use first time you need abbreviation it.

  1. 6. Circumstantial evidence has associated ITF with RCC progression into invasive tumors with poor clinical prognosis; however, to date little is known about the role that ITF playsin RCC pathophysiology [25]. Please check this sentence has written correctly. ITF if you use first time need abbreviation.

  1. As of 2021 there are about 15 molecules approved for the treatment of RCC 287 as single agents or within combinations by the European Medicines Agency (EMA) and the Federal Drug Administration (FDA)[29][30]. Please check this abbreviation written correctly.

  1. Line 177-179, Concurrently, interstitial fibroblasts, in the peritubular space, proliferate and deposit ECM when pushed towards a predominantly anaerobic metabolism. Please check this sentence has written correctly.

  1. Please write all the abbreviations after conclusion that you used throughout of the manuscript.
  2. Please make your conclusion more concise not elaborate.
  3. Please check your reference section and check all references has written according to journal guideline.

These are the few example of your manuscript.

General comments: A lot of typo and grammatical errors thought-out of the manuscript that should be revised must.

Author Response

Reviewer 2 comments and suggestions – rebuttal

Reply: We much appreciate the constructive comments to our manuscript entitled: ´´ Kidney Cancer and Chronic Kidney Disease: Too Close for Comfort´´. Our review has the objective t offer a wide-range perspective about commonalities observed between renal insufficiency and kidney cancer and prospective pharmacological therapies as well as biomarkers, under development, for both pathologies. In particular, we thank the reviewer for the constructive comments on manuscript design, absent references, abbreviations and the clarification of key mechanistic concepts concerning renal insufficiency and kidney cancer pathophysiology. We have updated our manuscript to reflect the reviewer´s considerations and we hope our corrections are satisfactory

The manuscript “Kidney Cancer and Chronic Kidney Disease: Too Close for 2 Comfort. The present review offers a perspective on the common biological traits shared between kidney cancer and chronic kidney disease and the available and prospective therapies for both conditions. Although I didn’t find any novelty in this review. In my opinion, this manuscript needs more insights information regarding kidney and chronic kidney disease and prospective therapies with molecular mechanism. Please see my comments and suggestions.

Comments and suggestions

  1. Keywords or keywords please check and confirm. And each keywords is small letter not capital.

Reply: All keywords now spelled in small letters.

  1. Post-glomerular filtration processes depend on an array of membrane-bound transporter proteins, expressed in both 41 the proximal and distal convoluted tubules, responsible for the removal of metabolic bi products and xenobiotics as well as the reabsorption of solutes, water, glucose, amino acids, and micro-nutrients. This sentence has written without citation.

Reply: The manuscript is updated with the following reference: Blaine et at 2015: ´´ Renal Control of Calcium, Phosphate, and Magnesium Homeostasis´´. (line 44)

  1. At the end of introduction you have to write aim of your current study that is missing.

Reply: The aim of our present review is now stated in the Introduction: ´´The aim of the present review is to offer a perspective about how the pathophysiological aspects shared by kidney cancer and chronic kidney disease can impact their diagnostics, the development of prospective therapies and the discovery of novel biomarkers´´ (lines 57-59)

  1. Section 2

The central mechanism in the cellular response to fluctuating O2 levels is the activity of the PHD-HIF axis. This interaction functions as a cellular O2 sensor, where Prolyl hy droxylases PHD consume intracellular O2 to catalyze the hydroxylation of nuclear 61 transcription factors known as Hypoxia-Inducible Factors HIFs. What is the meaning of this sentence? Please check and confirm. PHD-HIF if first time you have to must abbreviate it.

Reply: This statement refers to the cellular response to hypoxia. Prolyl hydroxylases (PHD) control the expression of Hypoxia-Inducible Factors (HIFs). When oxygen levels (O2) are normal, PHD hydroxylation of HIF is at steady-state and hydroxylated HIF are degraded. This process maintains HIF abundance at physiological levels. When a hypoxic event occurs O2 is low. This low O2 means that PHD can no longer hydroxylate the same amount of HIF as under normal conditions, therefore HIF abundance in the cells increases. This increase in HIF expression, since it is no longer degraded, is a key mechanism of cellular hypoxia response. HIF are transcription factors and the higher expression of HIF in hypoxia corresponds to higher expression of their target’s genes. The first mention of PHD-HIF in our manuscript is now spelled in full. (line 63)

  1. The activity of ATP-dependent membrane carriers is reduced and the expression of glycolytic enzymes is enhanced, in a push to preclude oxidative phosphorylation in the mitochondria in favor of non-O2 mediated an-78 aerobic metabolism ensuring ATP production. If you use first time you need abbreviation it.

          Reply: The first mention of adenosine triphosphate is now spelled in full (line 79).

  1. Circumstantial evidence has associated ITF with RCC progression into invasive tumors with poor clinical prognosis; however, to date little is known about the role that ITF plays in RCC pathophysiology [25]. Please check this sentence has written correctly. ITF if you use first time need abbreviation.

Reply: The sentence in question has been adapted to improve readability: ´´Circumstantial evidence shows an association between ITF and the progression of RCC into invasive tumors with poor clinical prognosis, however, to date little is known about the role that ITF plays in RCC pathophysiology´´ The first mention of intra-tumor fibrosis (ITF) in our manuscript is now spelled in full (lines 205-208).

  1. As of 2021 there are about 15 molecules approved for the treatment of RCC 287 as single agents or within combinations by the European Medicines Agency (EMA) and the Federal Drug Administration (FDA)[29][30]. Please check this abbreviation written correctly.

Reply: The wording and abbreviations in this sentence have been checked and confirmed.

  1. Line 177-179, Concurrently, interstitial fibroblasts, in the peritubular space, proliferate and deposit ECM when pushed towards a predominantly anaerobic metabolism. Please check this sentence has written correctly.

Reply: The sentence in question has been adapted to improve readability: ´´In a similar fashion, interstitial fibroblasts, in the peritubular space, can proliferate and deposit ECM when pushed towards a predominantly anaerobic metabolism´´ (lines 178-180)

  1. Please write all the abbreviations after conclusion that you used throughout of the manuscript.

Reply: All abbreviations used within the manuscript are now listed after the Conclusion section of our manuscript.

  1. Please make your conclusion more concise not elaborate.

Reply: The Conclusion section is now shorter in order to avoid redundancy and provide a concise summary of the key points in our manuscript. (lines 565-585)

  1. Please check your reference section and check all references has written according to journal guideline.

Reply: The reference section of our manuscript complies with the formatting guidelines of Biomedicines.  

These are the few example of your manuscript.

General comments: A lot of typo and grammatical errors thought-out of the manuscript that should be revised must.

Reply: Our manuscript has now been thoroughly spell checked in order to eliminate any typos and grammatical errors.

Round 2

Reviewer 2 Report

Authors significantly improved revised version than submitted version. Although, it needs some minor corrections before consider in this journal. 

1. Please check section 2. Regulating the oxygen supply to the kidneys 

The central mechanism in the cellular response to fluctuating O2 levels is the activity of the PHD-HIF (Prolyl hydroxylases Hypoxia-Inducible Factors) axis it has written correctly.

It will be Prolyl hydroxylases Hypoxia-Inducible Factors (PHD-HIF).

2. References section should be updated accordingly journal guideline.

3. Check typo and grammatical errors throughout of the manuscript.

Author Response

Reviewer 2 comments and suggestions – 2nd round – rebuttal

Reply: We once again thank the reviewer for the constructive comments. We have updated our manuscript accordingly. 

Comments and Suggestions for Authors

Authors significantly improved revised version than submitted version. Although, it needs some minor corrections before consider in this journal. 

  1. Please check section2. Regulating the oxygen supply to the kidneys 

The central mechanism in the cellular response to fluctuating O2 levels is the activity of the PHD-HIF (Prolyl hydroxylases Hypoxia-Inducible Factors) axis it has written correctly.

It will be Prolyl hydroxylases Hypoxia-Inducible Factors (PHD-HIF).

Reply: This change is now added to the manuscript (line 63)

  1. References section should be updated accordingly journal guideline.

Reply: The reference format was updated according to the journal guideline.

  1. Check typo and grammatical errors throughout of the manuscript.

Reply: The manuscript has been again proof-read for typos and grammatical errors

Submission Date: 19 October 2021

Date of this review: 17 Nov 2021 09:15:53